# LC/MS-Based Untargeted Metabolomics Study in Women with Nonalcoholic Steatohepatitis Associated with Morbid Obesity

**DOI:** 10.3390/ijms24129789

**Published:** 2023-06-06

**Authors:** Laia Bertran, Jordi Capellades, Sonia Abelló, Joan Durán-Bertran, Carmen Aguilar, Salomé Martinez, Fàtima Sabench, Xavier Correig, Oscar Yanes, Teresa Auguet, Cristóbal Richart

**Affiliations:** 1Grup de Recerca GEMMAIR (AGAUR)-Medicina Aplicada (URV), Departament de Medicina i Cirurgia, Universitat Rovira i Virgili, Institut d’Investigació Sanitària Pere Virgili, 43005 Tarragona, Spain; laia.bertran@fundacio.urv.cat (L.B.); joan.duran@urv.cat (J.D.-B.); carmenisabel.aguilar@urv.cat (C.A.); mariasalome.martinez@urv.cat (S.M.); fatima.sabench@urv.cat (F.S.); mariateresa.auguet@urv.cat (T.A.); 2Department of Electronic Engineering, Universitat Rovira i Virgili, Institut d’Investigació Sanitària Pere Virgili, 43007 Tarragona, Spain; jordi.capellades@iispv.cat (J.C.); xavier.correig@urv.cat (X.C.); oscar.yanes@urv.cat (O.Y.); 3Servei de Recursos Científics i Tècnics, Universitat Rovira i Virgili, 43007 Tarragona, Spain; sonia.abello@urv.cat; 4Unitat de Cirurgia, Facultad de Medicina i Ciències de la Salut, Hospital Universitari Sant Joan de Reus, Universitat Rovira i Virgili, Institut d’Investigació Sanitària Pere Virgili, 43204 Reus, Spain; 5CIBER de Diabetes y Enfermedades Metabólicas Asociadas, Instituto de Salud Carlos III, 28029 Madrid, Spain

**Keywords:** nonalcoholic fatty liver disease, nonalcoholic steatohepatitis, morbid obesity, metabolomics, lipidomics

## Abstract

This study investigated the importance of a metabolomic analysis in a complex disease such as nonalcoholic steatohepatitis (NASH) associated with obesity. Using an untargeted metabolomics technique, we studied blood metabolites in 216 morbidly obese women with liver histological diagnosis. A total of 172 patients were diagnosed with nonalcoholic fatty liver disease (NAFLD), and 44 were diagnosed with normal liver (NL). Patients with NAFLD were classified into simple steatosis (*n* = 66) and NASH (*n* = 106) categories. A comparative analysis of metabolites levels between NASH and NL demonstrated significant differences in lipid metabolites and derivatives, mainly from the phospholipid group. In NASH, there were increased levels of several phosphatidylinositols and phosphatidylethanolamines, as well as isolated metabolites such as diacylglycerol 34:1, lyso-phosphatidylethanolamine 20:3 and sphingomyelin 38:1. By contrast, there were decreased levels of acylcarnitines, sphingomyelins and linoleic acid. These findings may facilitate identification studies of the main pathogenic metabolic pathways related to NASH and may also have a possible applicability in a panel of metabolites to be used as biomarkers in future algorithms of the disease diagnosis and its follow-up. Further confirmatory studies in groups with different ages and sexes are necessary.

## 1. Introduction

Nonalcoholic fatty liver disease (NAFLD) has become one of the most prevalent chronic liver diseases in recent decades [1]. NAFLD is a metabolic disorder highly related to obesity and type 2 diabetes mellitus (T2DM) [2]. This disorder is a complex and progressive disease that can range from the benign condition called simple steatosis (SS) to its development into nonalcoholic steatohepatitis (NASH), an inflammatory-related state that sometimes leads to fibrosis and can evolve into hepatic cirrhosis and hepatocellular carcinoma [3]. Currently, there are still limited options for NAFLD clinical management in a specific manner due to the lack of noninvasive diagnostic tools and follow-up criteria for populations with a risk of progression [4].

Omics technologies are methods that can provide an accurate diagnosis and identify metabolites of a specific condition [5,6]. Their main advantage is that they can provide a huge amount of data in a very short period of time with an unbiased approach. The rapid evolution of artificial intelligence has also enabled the accurate analysis of large datasets produced by the omics sciences [7]. In relation to NAFLD, recent efforts have focused on metabolomic analyses-generating algorithms for the different histopathological stages of NAFLD. Different authors have reported several metabolites related to dysregulated pathways in NAFLD [8,9,10]. However, these studies presented heterogeneous cohorts in terms of age and sex with a reduced number of patients. Therefore, for a study of the relationship of these metabolites in NASH, it is necessary to expand these studies in larger and more homogeneous cohorts [11].

We recently performed an untargeted metabolomics analysis in morbidly obese (MO) women with associated T2DM [12]. Currently, to give continuity to this study, we plan to investigate another associated metabolic disorder, such as NAFLD/NASH. The aim of the present work was to define the metabolic profile of patients with NASH associated with obesity through liquid chromatography–mass spectrometry (LC/MS)-based untargeted metabolomics, identifying possible metabolites related to NASH metabolic pathways that allow us to have a better understanding of the physiopathology of this liver disease. For this proposal, we recruited a generic homogeneous cohort of women with associated NASH.

## 2. Results

### 2.1. Classification of the Participants

The LC/MS-based untargeted metabolomics analysis was assessed using serum samples from 216 MO women, which were classified according to their hepatic histology into normal liver (NL, *n* = 44), SS (*n* = 66) and NASH (*n* = 106) categories. First, the clinical and anthropometrical parameters of the studied cohort are expressed in Table 1.

In this regard, subjects were comparable in terms of age, body mass index (BMI) and waist–hip ratio. Analytically, there were also no differences in the levels of cholesterol, high-density lipoprotein–cholesterol (HDL-C) and low-density lipoprotein–cholesterol (LDL-C). In this sense, it should be considered that some subjects received prolonged treatment with lipid-lowering agents. The percentage of patients in each group treated with these drugs was 19.4% of NL, 41.5% of SS and 23.4% of NASH. The NAFLD patients (SS and NASH) presented significantly higher levels of glucose, glycated haemoglobin A1c (HbA1c), insulin, triglycerides, aspartate-amino transferase (AST), alanine-aminotransferase (ALT) and gamma-glutamil transferase (GGT) levels than the NL group. Moreover, SS subjects showed increased levels of lactate dehydrogenase (LDH) and ferritin levels compared to the NL group, but without significant differences vs. the NASH group.

### 2.2. Metabolic Profile in Morbidly Obese Women with NAFLD Compared to Those with NL

Subsequently, when we evaluated the metabolic profile in MO women with NAFLD in comparison with the NL group, we found significant differently concentrated metabolites in the family of lipids and derivatives (Table 2).

On the one hand, we found increased levels of diacylglycerols (DG 34:1 and DG 36:2), lyso-phosphatidylethanolamine (LPE) 20:3 and phosphatidylethanolamines (PE 34:1 and PE 40:6) in NAFLD compared to NL. On the other hand, we observed decreased levels of acylcarnitines. In addition, some species of phosphatidylcholines (PC 32:1, PC 32:2, PC 34:4 and PC 36:0) presented higher levels in NAFLD, while others (PC 34:1, PC O-32:1 and PC O-34-1) showed lower levels in NAFLD compared to the NL group.

### 2.3. Metabolic Profile in Morbidly Obese Women with SS Compared to Those with NL

Then, we wanted to evaluate in detail the metabolic profiles of the two stages of NAFLD separately in relation to that of those patients with a NL histology. Evaluating the metabolome of MO women with SS compared to those with NL, we also found significant differently concentrated metabolites in the family of lipids and derivatives (Table 3).

Regarding this comparison, we found increased levels of the PE 40:6 in SS subjects compared to NL, while we found lower levels of phosphatidylcholines in the SS group compared to the NL group.

### 2.4. Metabolic Profile in Morbidly Obese Women with NASH Compared to Those with NL

Later, we evaluated the metabolic profile of MO women with NASH compared to the NL group (Table 4).

Comparing MO women with NASH with those with NL, we found increased levels of DG 34:1, LPE 20:3, phosphatidylinositols and phosphatidylethanolamines. However, we found lower levels of acylcarnitines, linoleic acids and most of the sphingomyelins except for SM 38:1. In addition, we found that some phosphatidylcholines increased and others decreased in this analysis.

### 2.5. Metabolic Profile in Morbidly Obese Women with NASH Compared to Those with SS

Later, we wanted to elucidate which were the differential metabolites between MO women with NASH and those with SS (Table 5).

When we evaluated the differential metabolites between SS and NASH subjects, we found increased levels of DG 36:2, PC 36:0 and phosphatidylinositols in NASH subjects compared to SS subjects. On the contrary, we found decreased levels of PC 36:4 in the serum of NASH subjects compared to the SS group.

### 2.6. Heat Map of the Differently Concentrated Metabolites in the Studied Comparatives

To pool the results obtained, we have elaborated with a heat map showing the trends of these metabolites in the different studied comparatives (Figure 1).

In this graph, we can see that the NASH vs. NL comparison gives the most statistically significant metabolites. Moreover, we can observe that acylcarnitines, as well as sphingomyelins, monoacylglycerols, linoleic acid and some phosphatidylcholines, tend to be decreased in pathological conditions, while other metabolites, such as diacylglycerols, other phosphatidylcholines, phosphatidylinositols and phosphatidylethanolamines, tend to be increased in the disease.

### 2.7. Principal Component Analysis of the Metabolites Distribution in the Three Studied Groups

Finally, we performed a principal component analysis (PCA) to depict the multifactorial characteristics of NAFLD compared to the NL group (Figure 2). The main PCA components are not capable of separating the different patient groups; even with a total of 63% explained, there seems to be a tendency to separate NASH from the other two groups, but these are clearly overlapping. This means that even if the selected metabolites are individually significantly different between the groups, altogether they are not enough to model the differences between these three patient groups in this cohort.

## 3. Discussion

The novelty of this study lies in the fact that we performed an LC/MS-based untargeted metabolomics analysis in a well-characterized and homogeneous cohort of women with NAFLD and NASH associated with MO. In this work, we found a statistically significant difference in metabolite levels of lipid metabolites and derivatives. Our results identify a lipid profile in NASH.

In this regard, by analyzing metabolite families, we first reported decreased levels of acetyl-carnitine and hydroxybutyrylcarnitine in NASH patients vs. the NL group. There are few studies on these metabolites in NASH. Kalhan et al. found increased levels of acylcarnitines in patients with NASH, but with a limited cohort of only 24 patients, including both males and females [13]. In this sense, acylcarnitines are intermediate products of carnitine metabolism that have been associated with abnormal fatty acid oxidation [14,15]. The activity of fatty acid oxidation-involved enzymes can be evaluated using ratios of long-chain acetyl-carnitines/short-chain acylcarnitines [16]. It has been shown that long-chain acylcarnitines were strongly associated with NASH, while short-chain acylcarnitines, such as acetyl-carnitine and hydroxybutyrylcarnitine, did not show a clear pattern in this relationship [17]. However, Zhou et al. reported that a decline in medium- and short-chain acylcarnitines was associated with the severity of liver disease [18]. In this sense, the association between acylcarnitine metabolism and NASH is still uncertain and needs to be further studied.

Regarding sphingomyelins, we found increased levels of most of these metabolites, except for SM 38:1. Sphingomyelins are sphingolipids related to many biological functions, such as membrane structure [19,20]. Alterations in sphingolipid metabolism are related to insulin resistance and fatty liver [21,22]. Acid-sphingomyelinase (A-SMase) is the enzyme that catalyzes the lysosomal degradation of sphingomyelins into ceramides [23]. A-SMase has been reported to be upregulated in NAFLD and its deficiency has been related to the prevention of hepatic lipid accumulation. In this context, elevated ceramide levels, the product of sphingomyelins degradation, have been found in NASH [24]. However, the association of the sphingomyelins serum levels with NASH has been less studied. In this regard, in line with the results of this study, some authors also found decreased levels of SM 41:3, SM 36:0 and SM 38:0 in NASH [25,26].

In addition, we found decreased levels of monoacylglycerols (O-16:0) and increased levels of diacylglycerol 34:1 in serum samples of NASH subjects vs. the NL group. It has been seen that the deletion of the rate-limiting enzyme in the degradation of MG (monoacylglycerol lipase) ameliorates hepatic lipid accumulation and inflammation [27]. In this sense, an upregulation of this enzyme and the exacerbated consumption of monoacylglycerols may be related to NASH. On the other hand, diacylglycerol, which has been shown to be related to insulin resistance, is a lipid molecule that has been implicated in the generation and accumulation of lipid droplets [28]. It has been stated that increased levels of diacylglycerol are involved in the physiopathology of NAFLD and NASH [29]. In this sense, Vvedenskaya et al. also reported increased levels of DG 34:1 [25].

Moreover, we found decreased levels of linoleic acid in NASH subjects. Linoleic acid is an essential polyunsaturated fatty acid with a presumed insulin-sensitizing role [30,31]. In this sense, Puri et al. also reported notably decreased levels of linoleic acid in NASH [32]. Additionally, it has been shown that the circulating levels of linoleic acid from the diet have an inverse association with the risk of developing hepatic fibrosis [33]. Linoleic acid levels may be as low as that of an insulin sensitizer, which would imply a maintained persistence of hepatic insulin resistance throughout the course of the disease.

Regarding phospholipids, there are previous reports about the altered levels of phospholipids in NAFLD that can be related to a sustained disruption of lipid homeostasis [34,35,36]. In this regard, phosphatidylcholines and phosphatidylethanolamines are major components of cell membranes and are involved in a variety of cellular functions related to lipid metabolism [37,38]. Phosphatidylinositol is also a phospholipid that plays an important role in membrane structure and cell signaling [39]. It was reported that higher levels of phosphatidylinositol are associated with increased triacylglycerol and diacylglycerol accumulation in the liver and consequently, an induced lipotoxicity related to NASH [40]. However, the role of phosphatidylinositol in NASH pathogenesis has been poorly studied and remains quite uncertain [41,42]. In this study, we found increased levels of five phosphatidylinositol metabolites and four phosphatidylethanolamines species in NASH without any decrease in either family. In this regard, the literature agrees with our findings, given that Gorden et al. showed increased levels of PE 40:5, PE 40:6 and PI 36:1 in NASH subjects compared to the NL group [43]. Furthermore, regarding phosphatidylethanolamines, higher levels have previously been reported in NASH [36]. Increased levels of phosphatidylethanolamines were related to changes in lipogenesis, lipid oxidation, autophagy and apoptosis [44].

Moreover, we reported increased levels of lyso-phosphatidylethanolamine 20:3 in NASH subjects. In this sense, Tiwari-Heckler et al. agree with our findings, since the amount of lyso-phosphatidylethanolamines they found were significantly decreased in NAFLD/NASH patients compared to the NL group. They suggested that the decline in lyso-phosphatidylethanolamines is associated with the inability to further generate phosphatidylcholines and can lead to NAFLD progression [45].

On the other hand, we found increased levels of some phosphatidylcholines while observing decreased levels of others. It has been previously reported that phosphatidylcholine levels could be increased and decreased in NASH [37,46].

Finally, we would like to highlight that the increased levels of the phosphatidylcholine metabolites PC 32:2 and PC 36:0 and the phosphosethanolamine metabolites PE 34:2, PE 40:5 and PE 40:6 in our patients with NASH coincide with those of the patients with T2DM associated with MO in our previous study [12]. The only exception in these two groups is the PC 32:1, which showed lower levels in T2DM although it showed increased levels in NASH. These coincidences may indicate a significant relationship of phosphatidylcholines and phosphatidylethanolamines between the physiopathology of T2DM and NASH [47].

In summary, we have identified a characteristic lipid profile in NASH patients and, consequently, a likely pathogenic role of lipids in the pathophysiology of this disease. An environment with altered concentrations of diacylglycerides, sphingolipids, phospholipids, fatty acids and other metabolites can be related to the activation of endoplasmic reticulum stress [48]. Endoplasmic stress and oxidative stress play a role in the progression of NASH [49]. In addition, the phenomenon of lipotoxicity could be associated with organelle failure, cell damage and apoptosis and is linked to chronic inflammation [50,51]. The accumulated lipotoxicity, sustained insulin resistance, altered cell metabolic pathways and cellular necrosis, along with the activation of the innate immune system induced by the sustained metabolic disruption, may be the main drivers in the genesis of NASH.

This study has some limitations. First, although this is a homogeneous study, the results are only comparable with studies of adult women with morbid obesity, not with men, child or teen subjects or patients with other degrees of obesity. Then, we noted that some of our patients were in prolonged treatments with lipid-lowering agents, but we cannot control this parameter. Finally, we only found lipid metabolites that were differentially concentrated between groups, so the profile is focused on this group of metabolites in relation to previous studies.

All the same, it is important to highlight that this study was performed in a homogeneous cohort of MO women with NASH in an inflammation state without evidence of fibrosis. Therefore, we cannot comment on relation to fibrosis.

Although there are some personalized panels for the diagnosis of NASH currently on the market (NASH Steato Test 2, NASH Test 2, NASHnext, etc.) [52], most of them are based on biochemical parameters that can be modified by other processes, and these panels are unusually validated in cohorts of NASH not associated with other metabolic disorders, such as obesity or T2DM [53,54]. They can be used for guidance, but they possess low diagnostic specificity and few reliable correlations with the degree of evolution of the disease. Currently, the only reliable parameters are those obtained through the availability of histological studies. In this sense, the search for a new panel of biomarkers through metabolomic studies would allow us to obtain a specific diagnostic as well as a better definition of the evolutive degree of the process.

As was previously mentioned, these findings should be validated in other groups of patients, including those with wider ethnic variation and larger cohorts of different ages and sexes.

## 4. Materials and Methods

### 4.1. Participants

For this study, we selected 216 Caucasian women with MO (BMI > 40kg/m^2^) who underwent a laparoscopic bariatric surgery. Prior to the intervention, all patients were on a low-fat hypocaloric diet for at least one year. At the time of the intervention, a blood sample was collected, and a liver biopsy was indicated only by clinical criteria for diagnostic purposes. Informed consent approved by the Ethics Committee of Institut d’Investigació Sanitària Pere Virgili (IISPV) (CEIm; 23c/2015; 11 May 2015) was obtained in all cases. The study cohort belongs to the same group of patients studied in our previous metabolomics study elsewhere [12], so we included the schematic diagram of enrolled patients for both studies in Appendix A. Patients who had an acute illness, acute or chronic inflammatory or infective diseases or an end-stage malignant disease were excluded from this study. Menopausal women and women receiving contraceptive treatment were also excluded.

This cohort was composed of only women to evaluate a homogenous group to avoid the interference of sex, since men and women differ substantially regarding body composition, energy metabolism and hormones [55,56].

### 4.2. Hepatic Histopathological Classification

The liver samples were collected in a formaldehyde solution and, subsequently, biopsies were scored and classified by an experienced hepatopathologist through eosin-hematoxylin staining. In this sense, MO women with histological diagnoses were classified into NL (*n* = 44) and NAFLD (*n* = 172) groups. From these NAFLD subjects, two subgroups were established: SS (*n* = 66) and NASH (*n* = 106), according to Kleiner et al.’s criteria [57]. All cases of NASH corresponded to grade I-II with mild-moderate inflammation without the existence of hepatic fibrosis, stage F0 according to Ishak scores [58].

### 4.3. Anthropometrical and Biochemical Analysis

The anthropometrical evaluation included measurement of weight, height, waist-hip ratio and BMI calculation. Blood extraction was performed by specialized nurses through a BD Vacutainer^®^ system after overnight fasting and just before bariatric surgery. Venous blood samples were obtained in ethylenediaminetetraacetic acid tubes, which were separated into plasma and serum aliquots using centrifugation (3500 rpm, 4 °C, 15 min) and then stored at −80 °C until processing. The biochemical studies included glucose, insulin, HbA1c, total cholesterol, HDL-C, LDL-C, triglycerides, AST, ALT, GGT, alkaline phosphatase (ALP), LDH and ferritin levels, which were carried out using a conventional automated analyzer.

### 4.4. LC/MS Methods

The LC–MS method for the extraction and detection of metabolites in serum samples was described elsewhere [12].

### 4.5. Data Processing

Thermo.raw data files were transformed into .mzML files using Proteowizard’s MSconvert. Then, using R Software (version 4.1.2), .mzML files were processed using the RHermes R package to generate LC-MS/MS data of a preannotated list of detected ions. The XCMS R package was used to extract the quantification values from .mzML files. R software in-house scripts were used to integrate RHermes and XCMS packages outputs and to generate the data for statistics and plotting. Quantified ions with statistically significant results were manually identified using the previously acquired LC-MS/MS data to confirm their identity.

### 4.6. Statistical Analysis

All the values reported are expressed as medians and interquartile ranges. Differences between groups were calculated using the Mann–Whitney test. A *p*-value < 0.05 was considered to be statistically significant.

LC/MS data were processed using the HERMES R package previously described elsewhere [12,59]. Only SOI ions quantified in >80% of the samples were statistically tested for significant differences across the experimental groups using a one-way ANOVA. Statistical results were then adjusted using the false discovery rate (FDR) correction method. All the metabolites in the tables presented significant differences between groups (*p* < 0.05). Data of concentration were expressed in log fold change (FC) and p-values (Appendix A), as well as in the mean and standard deviation (Appendix A) of each metabolite in each group, as shown in Appendix A. R software (version 4.1.2) was used for the statistical analysis.

## 5. Conclusions

This LC/MS-based untargeted metabolomics study in a well-characterized and homogeneous cohort of MO women with NASH in an inflammatory state without fibrosis, has allowed us to identify a unique metabolic lipid profile associated with NASH. This profile includes increased levels of diacylglycerol, lyso-phosphatidylethanolamine, phosphatidylinositols and phosphatidylethanolamines associated with lower levels of acylcarnitines, linoleic acids and most of the sphingomyelins. This study needs to be further validated in larger cohorts with different age and sex groups.

## Figures and Tables

**Figure 1 ijms-24-09789-f001:**
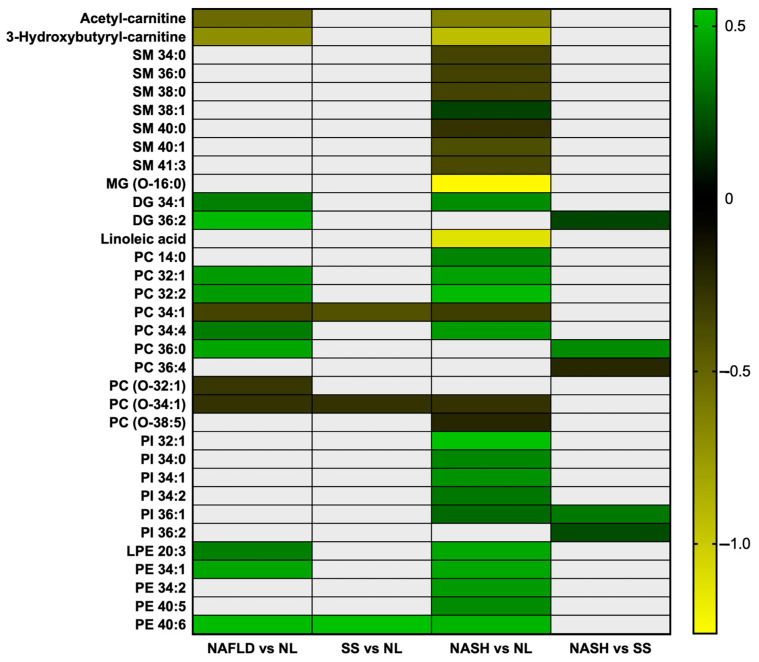
Heat map of the differently concentrated metabolites in the studied comparatives. NAFLD, nonalcoholic fatty liver disease; NL, normal liver; SS, simple steatosis; NASH, nonalcoholic steatohepatitis; SM, sphingomyelins; MG, monoacylglycerol; DG, diacylglycerol; PC, phosphatidylcholine; PI, phosphatidylinositol; LPE, lyso-phosphatidylethanolamine; PE, phosphatidylethanolamine. Green represents mostly higher levels and yellow represents mostly lower levels. This heat map was created using GraphPad Prism (version 7).

**Figure 2 ijms-24-09789-f002:**
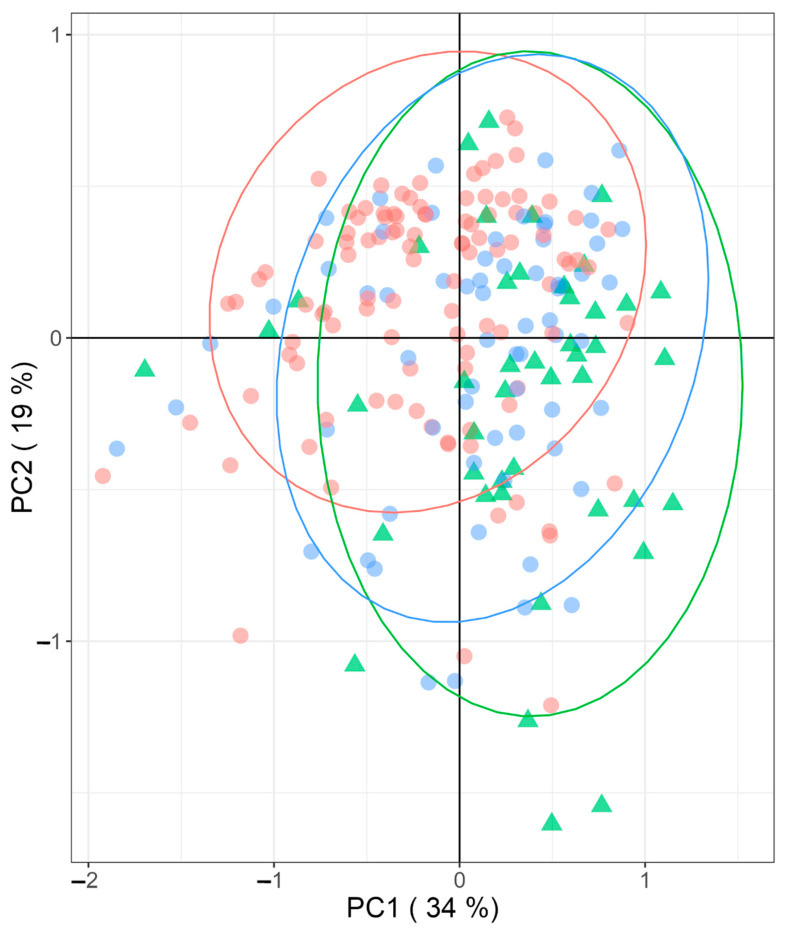
Principal component analysis (PCA) of significantly different concentrated metabolites between groups (NL (green triangles) and NAFLD (SS (blue circles) and NASH (red circles)); PC, principal component; NL, normal liver; SS, simple steatosis; NASH, nonalcoholic steatohepatitis; NAFLD, nonalcoholic fatty liver disease.

**Table 1 ijms-24-09789-t001:** Anthropometric and biochemical parameters of the study cohort.

Variables	NL (*n* = 44) Median (25–75th)	SS (*n* = 66) Median (25–75th)	NASH (*n* = 106) Median (25–75th)
Age (years)	46.47 (39.27–56.16)	47.68 (40.85–54.68)	48.74 (40.19–56.87)
BMI (kg/m^2^)	43.97 (41.64–49.38)	45.89 (43.01–51.49)	46.47 (43.31–50.55)
Waist-hip (m) ratio	0.89 (0.83–0.95)	0.93 (0.87–0.98)	0.92 (0.87–0.98)
Glucose (mg/dL)	90 (81–101)	109.50 (92.50–133.50) *	105 (90–132.50) *
HbA1c (%)	5.40 (5–5.70)	6.10 (5.47–7) *	5.80 (5.12–6.60) *
Insulin (mUI/L)	9.35 (5.67–13.07)	19 (11.14–33) *	16.34 (11.50–25.14) *
Cholesterol (mg/dL)	164 (140.50–200.25)	164.80 (144–192)	164.65 (146.50–186.75)
HDL-C (mg/dL)	39.50 (32.40–50.30)	36.75 (32–46)	38 (32.35–43)
LDL-C (mg/dL)	103.05 (79.32–127.35)	90.90 (76.25–114)	95 (76.17–116)
Triglycerides (mg/dL)	103.50 (77–135.50)	151 (116–197) *	146 (116–207) *
AST (UI/L)	22 (19–39)	35.35 (24.25–52.50) *	35 (25–54.25) *
ALT (UI/L)	23 (16–44.50)	35 (29–49) *	34 (25–58) *
GGT (UI/L)	17 (12–26.25)	26.25 (18.75–45.75) *	22.10 (15–50.75) *
ALP (Ul/L)	65 (52.50–76.50)	68 (54.50–76)	68 (58.80–78)
LDH (Ul/L)	388 (340–423.50)	427.50 (351.75–476.75) *	396.50 (344.25–481)
Ferritin (ng/mL)	36 (21.50–76.75)	75.14 (33.42–185.49) *	54 (27–119.30)

Data are expressed as medians and interquartile ranges. (*) Differences between NL and SS or NASH were considered significant when *p*-value < 0.05 using the Mann–Whitney test. NL, normal liver; SS, simple steatosis; NASH, nonalcoholic steatohepatitis; BMI, body mass index; HbA1c, glycosylated haemoglobin A1c; HDL-C, high density lipoprotein–cholesterol; LDL-C, low density lipoprotein–cholesterol; AST, aspartate-amino transferase; ALT, alanine-aminotransferase; GGT, gamma-glutamil transferase; ALP, alkaline phosphatase; LDH, lactate dehydrogenase.

**Table 2 ijms-24-09789-t002:** Metabolites with a significantly different concentration trend in the NAFLD group compared to the NL group.

Group	Class	Increased Levels	Decreased Levels
Lipids and derivatives	Acylcarnitines		Acetyl-carnitine, Hydroxybutyryl-carnitine
	Diacylglycerols (DG)	DG 34:1, DG 36:2	
	Lyso-phosphatidylethanolamines (LPE)	LPE 20:3	
	Phosphatidylcholines (PC)	PC 32:1, PC 32:2, PC 34:4, PC 36:0	PC 34:1, PC (O-32:1), PC (O-34:1)
	Phosphatidylethanolamines (PE)	PE 34:1, PE 40:6	

One-way ANOVA was used to identify significant differences. Data of concentration in log fold change (FC) and *p*-values, as well as the mean and standard deviation of each metabolite in each group, are given in Appendix A.

**Table 3 ijms-24-09789-t003:** Metabolites with a significantly different concentration trend in the SS group compared to the NL group.

Group	Class	Increased Levels	Decreased Levels
Lipids and derivatives	Phosphatidylcholines (PC)		PC 34:1, PC (O-34:1)
	Phosphatidylethanolamines (PE)	PE 40:6	

One-way ANOVA was used to identify significant differences. Data of concentration in log fold change (FC) and *p*-values, as well as the mean and standard deviation of each metabolite in each group, are given in Appendix A.

**Table 4 ijms-24-09789-t004:** Metabolites with a significantly different concentration trend in the NASH group compared to the NL group.

Group	Class	Increased Levels	Decreased Levels
Lipids and derivatives	Acylcarnitines		Acetyl-carnitine, Hydroxybutyryl-carnitine
	Sphingomyelins (SM)	SM 38:1	SM 34:0, SM 36:0, SM 38:0, SM 40:0, SM 40:1, SM 41:3
	Monoacylglycerols (MG)		MG (O-16:0)
	Diacylglycerols (DG)	DG 34:1	
	Fatty acids		Linoleic acid
	Lyso-phosphatidylethanolamines (LPE)	LPE 20:3	
	Phosphatidylcholines (PC)	PC 14:0, PC 32:1, PC 32:2, PC 34:4, PC 36:0	PC 34:1, PC 36:4, PC (O-32:1), PC (O-34:1), PC (O-38:5)
	Phosphatidylinositols (PI)	PI 32:1, PI 34:0, PI 34:1, PI 34:2, PI 36:1	
	Phosphatidylethanolamines (PE)	PE 34:1, PE 34:2, PE 40:5, PE 40:6	

One-way ANOVA was used to identify significant differences. Data of concentration in log fold change (FC) and *p*-values, as well as the mean and standard deviation of each metabolite in each group, are given in Appendix A.

**Table 5 ijms-24-09789-t005:** Metabolites with a significantly different concentration trend in the NASH group compared to the SS group.

Group	Class	Increased Levels	Decreased Levels
Lipids and derivatives	Diacylglycerols (DG)	DG 36:2	
	Phosphatidylcholines (PC)	PC 36:0	PC 36:4
	Phosphatidylinositols (PI)	PI 36:1, PI 36:2	

One-way ANOVA was used to identify significant differences. Data of concentration in log fold change (FC) and *p*-values, as well as the mean and standard deviation of each metabolite in each group, are given in Appendix A.

## Data Availability

Data are unavailable due to privacy and ethical restrictions.

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
