# Peer review of "LC/MS-Based Untargeted Metabolomics Study in Women with Nonalcoholic Steatohepatitis Associated with Morbid Obesity"

_ijms, 2023, doi:10.3390/ijms24129789_

Round 1

Reviewer 1 Report

Dear Authors,

The manuscript presents an interesting topic concerning the metabolomic profile of morbidity obese women in relation to nonalcoholic steatohepatitis (NASH). Using an untargeted metabolomics technique, The Authors showed altered levels of lipid metabolites in the studied population. The Authors suggested that the study's results could serve as biomarkers in future disease diagnosis and follow-up algorithms.

However, I have some considerations regarding missing elements in the manuscript.

  1. Please highlight the study's novelty at the begging of the discussion section.
  2. The number of patients analyzed in the previously published paper by Authors differs from those in the present manuscript. However, the Authors provided here the same bioethical approved NO. Is this population a part of a previous study? Please add the schematic diagram for enrolled patients in Supplementary Materials. 
  3. Authors should provide particular statistical tests used to obtain the presented p-values or other statistical indicators in the tables' footnotes (and descriptions of figures if applicable.) 
  4. The suggestion is to present the supplementary materials as two separate Tables in a text file instead of Excel sheets. Authors should replace commas in values with dots.
  5. The Excel sheet presents the p-value. Is the p-value after FDR correction? Please be precise about that. Please put the exact name of the statistical test used to compare values in the supplementary files.
  6. Please briefly explain the method for calculating a log fold change (FC) from the raw LC data in the Material and Methods section.
  7. The authors wrote that most of the subjects have been receiving lipid-lowering agents. Please precise it and provide a percentage of treated individuals for NL, SS, and NASH groups. Please discuss it briefly in the Discussion section.
  8. Please provide in the Materials and Methods section the name of the software applied in statistical evaluations or a language programming environment, if applicable.

The graphical representation of the metabolites abundance in studied groups, similar to the Figure from the authors' previous work, will be an advantage.

Reviewer 2 Report

The manuscript is an original article that assesses the lipid profile in women with nonalcoholic steatohepatitis (NASH) associated with morbid obesity using LC/MS-based untargeted metabolomics. The article is well-written in terms of the English language and the topic is of great interest to clinicians, nutritionists, and basic researchers. The paper provides a generous introduction and sufficient background to understand its message. 

However, there are a few major issues that should be addressed:

  1. Lines 43: The authors stated that “…..there are still limited options for NAFLD clinical management……due to the lack of noninvasive diagnostic tools and follow-up criteria”. The authors should know that there are non-invasive tests that are validated and  that use common blood parameters (e.g.: NASH Steato test 2, NASH Test 2, NASHnext). As such, please mention these aspects in the discussions and show the strengths of your findings and of the LC/MS technique versus these tests that are already commercially available for a long time.
  2. One of the aspects that could lead to a significant bias is the fact that it was not evaluated how many patients used lipid-lowering treatment and how many did not, in order to divide the cohort into subgroups according to this. Explain how this aspect could influence your findings.
  3. Nothing is mentioned in the article about the weaknesses of the research. Please insert a paragraph about this.
  4. The discussion chapter is poorly developed. Thus, although in the discussion chapter, the authors show their findings and compare them with the scientific flow, they do not provide any explanation of the increase or decrease of certain parameters of the lipid profile. Hypotheses should be issued on these findings which will later be validated through various studies.
  5. If the PCA analysis (Principal component analysis) showed that altogether lipid parameters that were individually identified as significantly increased or decreased cannot be used to differentiate between groups, how do you explain the utility of discovering this lipid profile from the point of view of clinical importance and implementation?

After the amendment of the above comments in the manuscript, I would be in favor of publishing this paper.

Minor editing of English language required.

Round 2

Reviewer 2 Report

Comments and Suggestions for Authors

The manuscript is an original article that assesses the lipid profile in women with nonalcoholic steatohepatitis (NASH) associated with morbid obesity using LC/MS-based untargeted metabolomics. I am glad that the authors made essential improvements to the manuscript, addressing the major points that I mentioned.

In this sense, the authors mentioned other existing and commercially available methods, about the factors that could have biased the study (e.g. lipid-lowering treatment) and brought explanations about the decreases/increases of the found parameters.

Significant portions of the discussion chapter were rewritten, thus bringing a better approach to the results obtained.

Also, a paragraph about the limitations of the study was added, thus completing the study.

In conclusion, the changes made to the article are significant and can support its publication in the journal.

Minor editing of English language required